# Structural Characteristics and In Vitro Digestibility of Malic Acid-Treated Corn Starch with Different pH Conditions

**DOI:** 10.3390/molecules24101900

**Published:** 2019-05-17

**Authors:** Chang Joo Lee, Jong Hee Na, Jun-Young Park, Pahn-Shick Chang

**Affiliations:** 1Department of Food Science and Biotechnology, Wonkwang University, Iksan 54538, Korea; cjlee@wku.ac.kr; 2Department of Biotechnology, College of Life Science and Biotechnology, Korea University, Seoul 02841, Korea; na843@kfri.re.kr; 3Department of Agricultural Biotechnology, Seoul National University, Seoul 08826, Korea; pjy1921@snu.ac.kr; 4Center for Food and Bioconvergence, Seoul National University, Seoul 08826, Korea; 5Research Institute of Agriculture and Life Sciences, Seoul National University, Seoul 08826, Korea

**Keywords:** malic acid, corn starch, modified starch, resistant starch, in vitro digestibility

## Abstract

The objective of this study was to investigate the influence of pH value on the in vitro digestibility of malic acid-treated corn starch in relation to its structural properties. Varying pH values (1.5–8.5) of 2 M malic acid solution were combined with corn starch in a forced-air oven at 130 °C for 12 h. Using Fourier-transform infrared spectroscopy (FT-IR), carbonyl groups were detected in malic acid-treated corn starch, indicating cross-linking through esterification. As the pH value of malic acid-treated corn starch decreased from 8.5 to 1.5, the resistant starch content increased from 18.2 to 74.8%. This was the result of an increased degree of substitution and was maintained after gelatinization. The granular structure of malic acid-treated corn starches was not destroyed, and the starches maintained birefringence. This malic acid-treated corn starch could be utilized in heat processed foods such as bread and cookies as well as in products with reduced calories.

## 1. Introduction

Consumer’s changing dietary preferences are increasing demand for low-calorie but high-satiety foods [1,2]. Starch is a major energy source and is found in the form of carbohydrates in many foods. Starch can be classified into three categories based on digestibility: rapidly digestible starch (RDS), slowly digestible starch (SDS), and resistant starch (RS) [3]. Resistant starch is normally non-digestible, and was recognized, academically, as a food product starting in the 1980s [4]. In general, starch with high amylose content can be considered as an ingredient for RS; however, if this starch is completely gelatinized it can become an ingredient in RDS. Given that starch delivered to the large intestine can be partially fermented by the gut microflora, RS is a dietary starch which is not digested in the small intestine [3]. Low-calorie processed foods manufactured using RS have better textures than those manufactured with dietary fiber [5]. Although foods manufactured with RS have similar total cholesterol content to those manufactured with dietary fiber; RS consumption has been shown to increase HDL cholesterol [6]. These various RSs are classified into the following four categories based on resistance to digestion: RS1 is physically inaccessible to digestion due to entrapment in a non-digestible matrix; RS2 is ungelatinized starch; RS3 is retrograded starch; and RS4 is chemically modified starch [4]. Production of RS4 is accomplished through chemical modifications, such as conversion, substitution, oxidation, acidizing, cross-linking, or a combination of these [7,8].

Multi-functional carboxylic acids including malic, tartaric, citric, and glutaric acid are generally employed for hydrogel synthesis and rheological characterization. Resistant starch content can be increased in corn starch through the application of citric acid and heat, resulting in esterification of hydroxyl groups in the starch molecule [9]. Unlike crystalline structure, cross-linking of starch is related to an increase in RS content [10]. Organic acids are reported to be less nutritionally harmful than inorganic acids and the digestion rate of the pancreas decreases with increasing degree of starch ester substitution [11,12]. Structural differences in the number of carboxyl groups affect the physicochemical characteristics of starch [11]. Malic acid is a C4-dicarboxylic acid, which comprises 69%–92% of the organic acids in grapes [13]. It is produced naturally by many organisms without negative nutritional impact, and DL-malic acid is a food-grade organic acid according to the Food Chemicals Codex (FCC) [14]. Additionally, malic acid has been used as an industrial food additive in the United States and European Union.

Corn (also known as maize) starch is a kind of cereal starches with distinctly low ash and protein contents. Its carbohydrate content is of high purity making it useful in several fields of industries. Corn starch is cheaper than other sources of starch; although, in its native form corn starch has limited industrial use due to its poor physical and mechanical properties including cohesive texture, heat, shear sensitivity, lack of clarity, and retrogradation or precipitation upon storage. Native corn starch is chemically and physically modified for general use [15]. Corn starch has garnered attention due to its wide range of dietary and industrial applications. Extensive research is required to enhance our understanding of the potential chemical, physical, and biotechnological modification possible for corn starch. The purpose of this study was to prepare corn starch with a high contend of RS using malic acid under varying pH conditions, followed by investigation of the physicochemical properties, structural characteristics, and in vitro digestibility of the resulting product.

## 2. Results and Discussion

### 2.1. Light Microscope

The granularity of the malic acid-treated corn starches, arranged according to their pH value, are shown in Figure 1. A control sample and a malic acid-treated sample displayed various starch shapes: round, half elliptical, elliptical spherical, and round polygonal [16,17].

Increased temperature did not affect the shape or size of the native corn granules. Starch granules treated with 2.0 M malic acid were not destroyed [18]. Corn starch granules did not vary significantly in size with pH value of solution and maintained cohesion. Granule shape of malic acid-treated starch was not destroyed by any concentration of malic acid. The Maltese cross was observed; in addition to the semi-crystalline structure and radially ordered alignment of amylose and amylopectin [19]. The Maltese cross was clearly observable in all starch samples, indicating that the internal structure of the starch remained regularly arranged with changing pH value (Figure 1A_2_–H_2_). Thus, the shape and structure of starch granules did not change in response to heat or malic acid.

### 2.2. Degree of Substitution (DS)

The degree of malic acid substitution (DS) of malic acid-treated corn starch is shown in Table 1. There was no bond formation in either the heat-treated or control corn starch, the value of malic acid substitution and the DS was from 0.0002 to 0.0004. The DS value of the control (pH 1.5) without malic acid treatment was 0.0383 at 130 °C. The DS of malic acid-treated corn starch decreased from 0.2156 to 0.0349 as the pH increased.

Malic acid has two carboxyl groups which are able to form two ester bonds. The DS may be affected by the number of carboxyl groups belonging to an organic acid; however, steric hindrance can interrupt ester bond formation. Substitutions and bridge reactions occur more frequently in amorphous areas of starch molecules [20]. This suggests DS increases as pH decreases due to the destruction of some crystalline regions and the consequent increase in amorphous areas.

### 2.3. Fourier-Transform Infrared Spectroscopy (FT-IR)

Fourier-transform infrared spectroscopy (FT-IR) was used to evaluate the structural characteristics of the functional groups in the starches. Peaks in the range of 3,000–3,500 cm^−1^ represent the presence of hydroxyl groups (-OH), and the peak at 2,930 cm^−1^ represent C-H bond stretching [21]. Peaks in the range of 980–1,240 cm^−1^ may be caused by the stretching vibration of C-O bond [22]. The peak near 1,730 cm^−1^ is a carbonyl group (C=O) [18,23] and the malic acid-treated corn starches at pH values of 1.5, 3.5, and 5.5 showed the carbonyl peaks at 1,722 cm^−1^, especially high carbonyl peak intensity at pH 1.5 (Figure 2). None of the samples demonstrated a carbonyl peak at pH 7.0 or pH 8.5. This is due to the destruction of intermolecular hydrogen bonds at lower pH values, indicating a bridge formation by ester bond between starch and malic acid. As the peak intensity of malic acid-treated starch samples increased, the contents of RS also increased (*p* < 0.05). The RS content was: 87.9% for pH 1.5 MT, 51.9% for pH 2.5 MT, and 46.4%, for pH 5.5 MT. Thus, the condition of high pH caused hydrolysis of the bridge bonds in malic acid-treated corn starches [18].

### 2.4. X-ray Diffraction

The X-ray diffraction patterns of malic acid-treated corn starches are shown in Figure 3. The internal order of a starch granule is demonstrated by X-ray diffraction patterns A, B, and C types [24]. All samples showed the typical A-type diffraction pattern of two peaks at 17°, and one peak at 23° at angle 2θ. Both shorter double helices and inner crystallites in A-type starches were more easily digestible, containing large amount of RDS and SDS when compared to B-type starches, which contain large amount of RS [25]. It was previously reported that heat and acid treatment of corn starches did not affect the crystalline structure or x-ray diffraction patterns of A-type [26]. X-ray diffraction patterns remained unchanged in the malic acid-treated corn starches, but crystallinity and peak intensity decreased with decreased pH value. Hydrogen bonds are known to stabilize a double helix. However, when ester bonds are substituted relative crystallinity of the double helix is reduced.

### 2.5. In Vitro Digestibility

Content analysis of starches and their digestive characteristics (RDS, SDS, and RS) are shown in Table 2. For each native corn starch and heat-treated corn starch RDS was 8.2–8.6%, SDS was 67.1–68.4%, and RS was 23.0–24.7%. There was no significant difference between the samples. However, starches treated with heat and malic acid showed an RDS increase from 9.1% to 46.2%, an SDS increase from 3.0% to 11.7%, and an RS decrease from 87.9% to 46.4%; as the pH values increased from 1.5 to 5.5. Additionally, there were no significant differences in digestion rates of RDS, SDS, and RS between pH value 7.0 and 8.5. The content of RDS was the highest in malic acid-treated corn starch at pH 5.5. The RDS content increased until a pH of 5.5, and then decreased 22.4%, as pH increased to 7.0. This suggests RDS content increased due to amylose destruction and amylopectin structure, when acid and heat was applied, and remained parts of starch that can form ester bonds with malic acid became RS.

The more starch forming bridge bonds, the harder it became for α-amylase to infiltrate through porous starch, which led to greater resistance to digestion [27]. The malic acid-treated corn starches showed resistance against infiltration by α-amylase due to their internal ester bonds between malic acid and starch molecules, which is caused by increased DS. However, if the cross-linking completely prevented the infiltration of α-amylase, an increase in RDS would not have been possible. Interference with cross-linking in the α-amylase starch complex may be a factor in RS increase [11,28]. Despite the diffusion of α-amylase after infiltration of granules, bridge bonds in the starch molecule resisted digestion. This was previously reported to be due to the starch granules bonding with the citric acid, resisting the swelling of the starch granules [9].

When starch is heated during cooking, which is common, RS can decrease due to destruction of ester bonds. Table 2 shows the digestibility characteristics for the starches analyzed in this study. The malic acid-treated corn starch at pH 1.5 was 74.8% RS which was an 87.9% decrease from its pre-cooked state. Even though other malic acid-treated starches showed decreases in RS content, there was a strong correlation with DS. Substituted starch and DS of RS fraction displayed high relation to each other (r = 0.881, *p* < 0.05). The RS of the control sample (pH 1.5) decreased from 25.7% to 21.9% when cooked. Lower pH values for malic acid-treated samples corresponded to higher percentages of RS and increased heat stability.

## 3. Materials and Methods

### 3.1. Materials

Normal corn starch (~25% amylose) was obtained from Samyang Genex (Incheon, South Korea). DL-malic acid was purchased from Sigma (M1210, Sigma-Aldrich, St. Louis, MO, USA). The enzymes used in the starch digestion were porcine pancreatin (P7545, activity 8 × USP/g, Sigma, St. Louis, MO, USA) and amyloglucosidase (AMG 300L, activity 300 AGU/mL, Novozymes Inc., Bagsvaerd, Denmark), where USP and AGU stand for United States Pharmacopia and amyloglucosidase activity, respectively. All chemicals and reagents used were of analytical grade.

### 3.2. Preparation of Malic Acid-Treated Corn Starch

The acidities of malic acids (2.0 M) with the various pH values (3.5, 5.5, 7.0, or 8.5) were adjusted using 10 M sodium hydroxide (NaOH). Malic acid with pH 1.5 was made without addition of NaOH. Corn starch (20 g) and acid malic solution of corrected pH (20 mL) were combined to form a solution and mixed in stainless-steel container for 16 h at room temperature. The bowls were placed in an air drying oven at 40 °C for 24 h to a uniform moisture concentration (~10%). The dried mixture was ground and either placed in an air drying oven at 130 °C or left at room temperature, for 12 h. Next, the mixture was thoroughly washed with distilled water to remove unreacted malic acid and 95% ethanol. The malic acid-treated corn starch was then dried in an air drying oven at 40 °C before being ground. A control sample of starch was prepared following the same procedure, but not heated.

### 3.3. Light Microscopy

Malic acid-treated corn starches were observed with a light microscope (Olympus BX40, Olympus Optical Co., Ltd., Japan) with and without polarizing plate. Glycerol was used to disperse all samples on a glass slide and to reduce air bubbles.

### 3.4. Fourier-Transform Infrared Spectroscopy (FT-IR)

To investigate FT-IR spectra of starches, FT-IR (Vertex 80V; Bruker, CA, USA) was used. Spectra were measured ranging from 4,000 to 600 cm^−1^ in transmission mode at a resolution of 4 cm^−1^. Samples were diluted with KBr (1:100, *v/v*) before acquisition.

### 3.5. Degree of Substitution (DS)

Degree of substitution was measured to investigate the average number of malic acid substituted hydroxyl groups per anhydroglucose unit of starch. The value was measured following the method of Miladinov with slight modifications [29]. Malic acid-treated corn starch (0.5 g) was placed in a 250 mL-glass beaker with 50 mL of distilled water. The pH value was measured after stirring for 1 h at 30 °C. Each beaker received 25 mL of 0.5 N NaOH to release the substituted groups from the malic acid-treated starch and the solution was stirred for 24 h at 50 °C. The excess NaOH was titrated back to original pH with 0.05 N hydrochloric acid (HCl). The DS value was calculated as follows: $$\mathrm{DS}=\frac{{162\times (N}_{\mathrm{NaOH}}{\times V}_{\mathrm{NaOH}}{-N}_{\mathrm{HCl}}{\times V}_{\mathrm{HCl}})}{1,000\times W-116.19{\times (N}_{\mathrm{NaOH}}{\times V}_{\mathrm{NaOH}}{-N}_{\mathrm{HCl}}{\times V}_{\mathrm{HCl}})}$$
where, DS is degree of substitution, W is the sample weight (g), N_NaOH_ is the normality of NaOH, V_NaOH_ is the volume of NaOH, N_HCl_ is the normality of HCl used to back titrate, and V_HCl_ is the volume of HCl used for back titration.

### 3.6. In Vitro Digestibility

Starch digestibility was analyzed using the Englyst’s method [4] with slight modifications [30]. To prepare the enzyme solutions, porcine pancreatin (2 g, dry basis, db) was added to 24 mL of distilled water in a 50-mL glass beaker and stirred for 10 min. The solution was then centrifuged at 1,500× *g*, for 10 min at 4 °C, to obtain a cloudy supernatant. The supernatant (20 mL) was mixed with 0.4 mL of amyloglucosidase and 3.6 mL of distilled water. A 0.75 mL measurement of sodium acetate buffer (0.1 M, pH 5.2) and a glass bead were added to a 2 mL microtube with a 30 mg starch sample, and either cooked for 10 min or not cooked. After cooling the microtube to 37 °C, 0.75 mL of the enzyme solution was added and incubated in a shaking incubator (240 rpm). The microtubes were taken out after 20 and 240 min, boiled for 10 min in a heating block to stop the reaction, and cooled to room temperature. The microtubes were centrifuged at 5000× *g*, for 10 min at 4 °C. The amount of glucose in the supernatant was used to measure by the GOD-POD kit (Embiel Co., Gunpo, Korea). The amount of glucose after 20 min of enzyme reaction at 37 °C indicated RDS, and that obtained after incubation for 20-240 min was SDS. The RS was the starch not hydrolyzed after 240 min incubation.

### 3.7. X-ray Diffraction

X-ray diffraction patterns of starches were investigated using an X-ray diffractometer (Model D8 Advance, Bruker, Karlsruhe, Germany) which was operated at 40 kV and 40 mA producing CuKα radiation of 1.54 Å wavelength, and scanned through the 2θ range of 3−30° with step time of 0.5 sec.

### 3.8. Statistical Analysis

Experiments were carried out in triplicate, and the mean values and standard deviations were presented. Analysis of variance was performed, and Duncan’s multiple range test was used for assessment of significant differences. Statistical analyses were performed using SPSS for Windows 18.0 software (SPSS Inc., Chicago, IL, USA).

## 4. Conclusions

To compare the in vitro digestibility, physicochemical, and structural characteristics of malic acid-treated corn starch samples, pH was varied from 1.5–8.5 before samples were heated to 130 °C for 12 h. The RS content of the malic acid-treated samples increased as the pH value decreased. The DS values and the FT-IR spectra revealed ester bond formation between the malic acid and starch. By lowering pH value, in addition to heating, RS content continued to increase even after gelatinization, for the malic acid treated samples. In conclusion, high RS content is possible, with low pH (1.5) malic acid treatment. These conditions lead to the desired level of substitution. The increased RS content may have resulted from cross-linking. After malic acid and heat treatments, the internal structure of the corn starch became heat-stable. These results suggest that pH affects esterification of malic acid impacting in vitro digestibility, physicochemical, and structural characteristics of corn starch. The malic acid-treated corn starch had high RS content and heat stability, and is therefore suitable for heat-treated food, such as bread or cookie, in addition to low-calorie foods. This heat stable RS formation technology could be applied in daily lives of people while motivating further research.

## Figures and Tables

**Figure 1 molecules-24-01900-f001:**
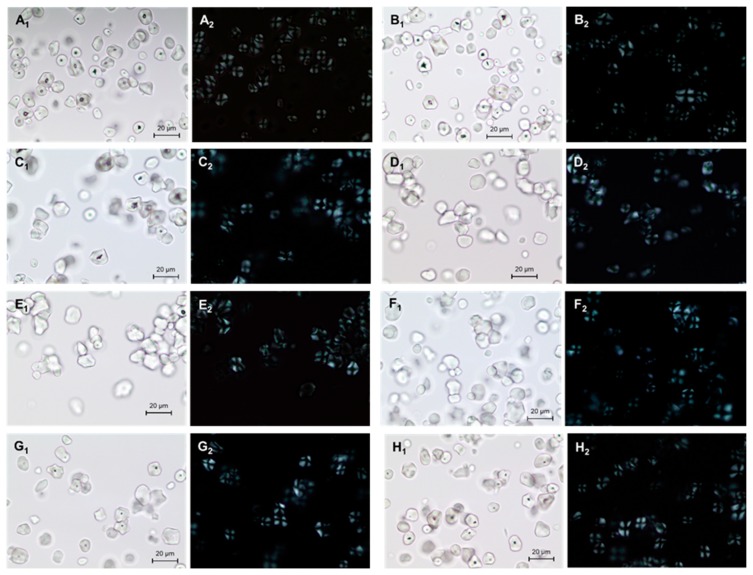
Light micrographs of granules of native and malic acid-treated corn starch: (**A**) native corn starch, (**B**) corn starch with 130 °C treatment, (**C**) corn starch with pH 1.5 treatment (control), (**D**) corn starch with pH 1.5 MT, (**E**) corn starch with pH 3.5 MT, (**F**) corn starch with pH 5.5 MT, (**G**) corn starch with pH 7.0 MT (**H**) corn starch with pH 8.5 MT, (1) light micrographs (2) under polarized light. *MT, malic acid treatment.

**Figure 2 molecules-24-01900-f002:**
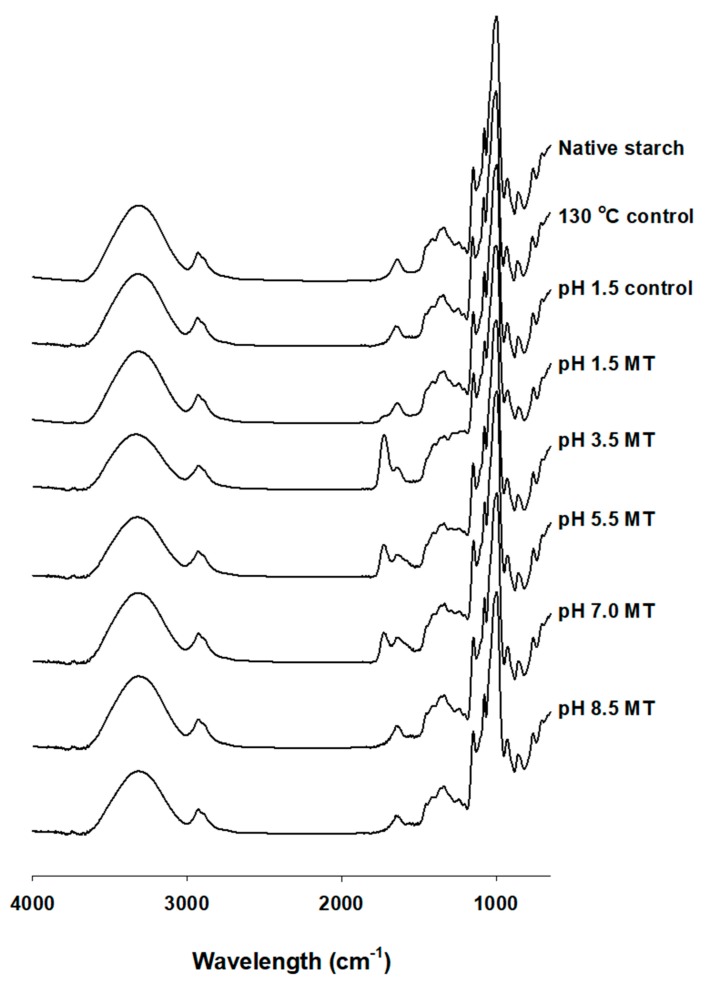
Fourier-transform infrared spectroscopy (FT-IR) spectra of native and malic acid-treated corn starches. MT, malic acid treatment.

**Figure 3 molecules-24-01900-f003:**
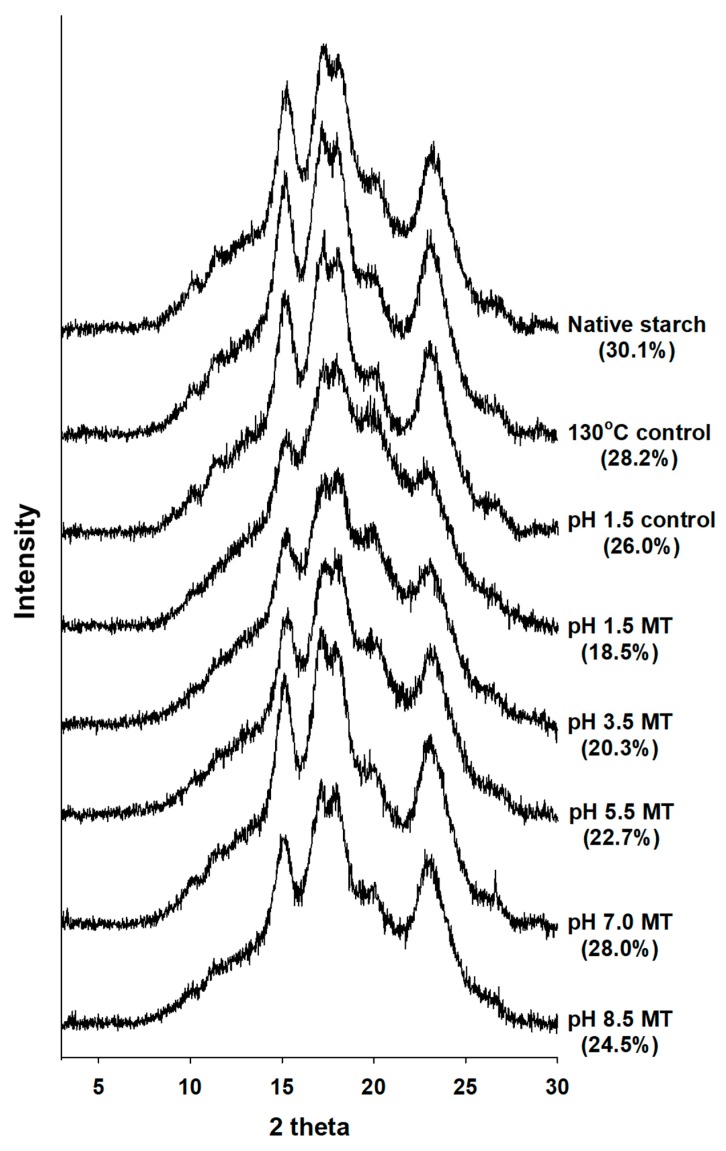
X-ray diffraction patterns and relative crystallinity of native and malic acid-treated corn starches. Numbers in parentheses indicate the percentage of crystallinity. MT, malic acid treatment.

**Table 1 molecules-24-01900-t001:** Degree of malic acid substitution of native and malic acid-treated corn starches.

Sample	Degree of Substitution
Native starch	0.0004 ± 0.0002 ^a^
130 °C control	0.0002 ± 0.0001 ^a^
pH 1.5 control	0.0383 ± 0.0067 ^b^
pH 1.5 MT	0.2156 ± 0.0103 ^c^
pH 3.5 MT	0.1979 ± 0.0035 ^d^
pH 5.5 MT	0.0897 ± 0.0064 ^e^
pH 7.0 MT	0.0366 ± 0.0071 ^b^
pH 8.5 MT	0.0349 ± 0.0032 ^b^

^a–e^ The values with different superscripts within a column are significantly different (*p* < 0.05) by Duncan’s multiple range test. MT, malic acid treatment.

**Table 2 molecules-24-01900-t002:** In vitro digestibility characteristics of native and malic acid-treated corn starches.

Sample	Uncooked Starch *			Cooked Starch		
	RDS (%)	SDS (%)	RS (%)	RDS (%)	SDS (%)	RS (%)
Native starch	8.20 ± 0.73 ^a^	67.1 ± 1.29 ^f^	24.7 ± 1.99 ^a^	72.5 ± 0.39 ^d^	8.47 ± 0.59 ^a^	19.1 ± 0.69 ^b^
130 °C control	8.62 ± 2.06 ^a^	68.4 ± 1.78 ^f^	23.0 ± 1.92 ^a^	76.8 ± 1.07 ^e^	6.70 ± 0.34 ^a^	16.5 ± 0.76 ^a^
pH 1.5 control	29.0 ± 1.45 ^c^	45.2 ± 1.78 ^d^	25.7 ± 0.83 ^ab^	71.6 ± 1.60 ^d^	6.52 ± 0.99 ^a^	21.9 ± 0.69 ^c^
pH 1.5 MT	9.12 ± 0.68 ^a^	3.01 ± 0.53 ^a^	87.9 ± 1.21 ^f^	17.4 ± 1.78 ^a^	7.83 ± 1.02 ^a^	74.8 ± 1.34 ^f^
pH 3.5 MT	36.7 ± 0.50 ^d^	11.7 ± 0.66 ^c^	51.9 ± 1.57 ^e^	38.6 ± 1.99 ^b^	12.6 ± 0.94 ^b^	48.8 ± 1.05 ^e^
pH 5.5 MT	46.2 ± 1.06 ^e^	7.54 ± 1.32 ^b^	46.4 ± 1.09 ^d^	49.0 ± 1.89 ^c^	4.32 ± 0.77 ^a^	46.8± 1.15 ^d^
pH 7.0 MT	23.0 ± 1.66 ^b^	47.4 ± 1.36 ^de^	29.6 ± 1.71 ^c^	70.9 ± 1.17 ^d^	7.64 ± 2.14 ^a^	21.5 ± 1.25 ^c^
pH 8.5 MT	22.4 ± 1.22 ^b^	48.9 ± 1.44 ^e^	28.6 ± 0.26 ^b^	74.0 ± 1.59 ^de^	7.88 ± 1.64 ^a^	18.2 ± 0.63 ^ab^

^a–f^ The values with different superscripts within a column are significantly different (*p* < 0.05) by Duncan’s multiple range test. * RDS, rapidly digestible starch; SDS, slowly digestible starch; RS, resistant starch; MT, malic acid treatment.

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
