# Peer review of "Structural Characteristics and In Vitro Digestibility of Malic Acid-Treated Corn Starch with Different pH Conditions"

_molecules, 2019, doi:10.3390/molecules24101900_

Round 1

Reviewer 1 Report

The presented article addresses the chemical characteristics and in vitro digestibility of maize starch treated with malic acid, and provides data of interest that deserve to be published in Molecules.

However, some considerations should be taken into account:

- In tables 1 and 2, the numerical values are accompanied by letters whose meaning must be detailed.

- Detail that the dilution of AM 2M of pH 1.5 does not require the addition of NaOH (line 225)

- In line 226, "malic acid (20 ml)" should be changed to "acid malic solution of corrected pH (20 ml)".

- Bibliographic references 4, 6 and 7 should be eliminated because they are very collateral with the article.

- But the most important thing is that, in the conclusions, the industrial / economic utility of treating each 20 g of corn starch with 5.36 g of AM should be explained.

Author Response

Journal: Molecules

Manuscript ID: molecules-503337

Title: Structural Characteristics and In Vitro Digestibility of Malic Acid-Treated Corn Starch with Different pH Conditions

Responses to reviewer’s comments

: We are very grateful for the thorough review with informative comments and constructive suggestions by the reviewers. We considered the reviewer’s comments with meticulous care. We have prepared answers to the reviewer’s comments as sufficiently as possible, and corrections and clarification in our revised manuscript have been made by rewriting related sentences or paragraphs, considering the reviewer’s suggestions. The detailed responses are listed below.

•       Reviewer #1

The presented article addresses the chemical characteristics and in vitro digestibility of maize starch treated with malic acid, and provides data of interest that deserve to be published in Molecules. However, some considerations should be taken into account.

1. In tables 1 and 2, the numerical values are accompanied by letters whose meaning must be detailed.

: We considered the reviewer’s comment with meticulous care. However, we could not find the problems on the description of superscripts in table 1 and 2. Statistical information of different superscripts is already appended to the footnotes of the table.

2. Detail that the dilution of AM 2M of pH 1.5 does not require the addition of NaOH (line 225). In line 226, "malic acid (20 ml)" should be changed to "acid malic solution of corrected pH (20 ml)".

: We apologize for the ambiguous information on the Methods & Materials in our manuscript. We have rectified the sentences in order to avoid ambiguity as reviewer suggested (line 221-223 in the revised manuscript).

→ “The acidities of malic acids (2.0 M) with the various pH values were adjusted using 10 M sodium hydroxide (NaOH). Corn starch (20 g) and acid malic solution of corrected pH (20 mL) were combined to form a solution and mixed in stainless-steel container for 16 h at room temperature.”

3. Bibliographic references 4, 6 and 7 should be eliminated because they are very collateral with the article.

: As the reviewer suggested, we have deleted those references and re-order the references in the revised manuscript.

4. But the most important thing is that, in the conclusions, the industrial / economic utility of treating each 20 g of corn starch with 5.36 g of AM should be explained.

: We agree with the reviewer’s comment on this and apologize for the incomplete description. For clarification of our intent in the Conclusions, we have rewritten the sentence (line 210-211 in the revised manuscript).

“This heat stable RS formation technology could be applied in daily lives of people while motivating further research.”

Reviewer 2 Report

The authors investigate the structural properties and in vitro digestion of malic acid-treated corn starch with different pH conditions. The results provide some information for producing corn starch with high RS content using malic acid. I have some comments as the below.

1. More information is needed for corn starch, for example, the contents of amylose, protein and lipids in starch.

2. How to prepare the sample of pH 1.5 control?

3. Fig. 1. It is better to improve the concentration of starch. The starch granules are too few to exhibit their morphology and size clearly.

4. L85. For size, it is difficult to obtain the information from Fig. 1. It is better to analyze the granule size using a laser diffraction instrument.

5. Table 1. For the very low value, it is better to use the unit of 10-3.

6. Fig. 2. Usually, the wavelength is from 4000 to 1000 cm-1, not from 1000 to 4000 cm-1.

Author Response

Journal: Molecules

Manuscript ID: molecules-503337

Title: Structural Characteristics and In Vitro Digestibility of Malic Acid-Treated Corn Starch with Different pH Conditions

Responses to reviewer’s comments

: We are very grateful for the thorough review with informative comments and constructive suggestions by the reviewers. We considered the reviewer’s comments with meticulous care. We have prepared answers to the reviewer’s comments as sufficiently as possible, and corrections and clarification in our revised manuscript have been made by rewriting related sentences or paragraphs, considering the reviewer’s suggestions. The detailed responses are listed below.

•       Reviewer #2

The authors investigate the structural properties and in vitro digestion of malic acid-treated corn starch with different pH conditions. The results provide some information for producing corn starch with high RS content using malic acid. I have some comments as the below.

1. More information is needed for corn starch, for example, the contents of amylose, protein and lipids in starch.

: We appreciate the reviewer’s informative comment strengthening the quality of our manuscript and totally agree with this. We have included the additional information about starch used in this study (line 214 in the revised manuscript).

“Normal corn starch (~25% amylose) was obtained from Samyang Genex (Incheon, South Korea).”

2. How to prepare the sample of pH 1.5 control?

: We already described the information about preparation for the control sample in M&M section. Corn starch (20 g) and acid malic solution of pH 1.5 (20 mL) were combined to form a solution and mixed in stainless-steel container for 16 h at room temperature and then dried at 40 °C in air drying oven. A control sample of starch was prepared following the same procedure, but without heating.

3-4. Fig. 1. It is better to improve the concentration of starch. The starch granules are too few to exhibit their morphology and size clearly. L85. For size, it is difficult to obtain the information from Fig. 1. It is better to analyze the granule size using a laser diffraction instrument.

: We totally agree with the reviewer’s comment on this. Therefore, we have changed the data set in the Fig. 1 in order to exhibit the morphology and size of starch clearly. Please see the Fig. 1 in the revised manuscript.

5. Table 1. For the very low value, it is better to use the unit of 10-3.

: We appreciate the reviewer’s comment. However, the degree of substitution is generally expressed as being in our manuscript.

6. Fig. 2. Usually, the wavelength is from 4000 to 1000 cm-1, not from 1000 to 4000 cm-1.

: We totally agree with the reviewer’s comment on this. Therefore, we have changed the Fig. 2 as reviewer suggested. Please see the Fig. 2 in the revised manuscript.

Round 2

Reviewer 1 Report

The article presented on the digestibility of maize treated with malic acid at different pHs provides new data regarding the state of the art and deserves to be published in Molecules, although some aspects should be revised:

1.- Tables 1 and 2 show letters as supraindices whose meaning is unknown.

2.- In section 4.2, it must be specified that the 2M malic acid solution does not require the addition of NaOH to reach the pH = 1.5

3.- Bibliographic citations 4, 6 and 7 should be eliminated because they are superfluous in relation to the article.

Author Response

Journal: Molecules

Manuscript ID: molecules-503337

Title: Structural Characteristics and In Vitro Digestibility of Malic Acid-Treated Corn Starch with Different pH Conditions

Responses to reviewer’s comments (Round 2)

: We are very grateful for the thorough review with informative comments and the constructive suggestions by the reviewers. We considered the reviewer’s comments with meticulous care. We have prepared answers to the reviewer’s comments as sufficiently as possible. The detailed responses are listed below. In addition, we used “Track Changes” for clear visualization of changes in the manuscript.

       Reviewer #1

The presented article addresses the chemical characteristics and in vitro digestibility of maize starch treated with malic acid, and provides data of interest that deserve to be published in Molecules. However, some considerations should be taken into account.

1. Tables 1 and 2 show letters as supraindices whose meaning is unknown.

: Statistical information of different superscripts is already appended to the footnotes of table; however, we have revised the Tables and their footnotes to convey what we stand for properly. Please see the footnotes in the Table 1 and 2.

2. In section 4.2, it must be specified that the 2M malic acid solution does not require the addition of NaOH to reach the pH = 1.5.

: We apologize for the ambiguous information on the Methods & Materials in our manuscript. We have rectified the sentences in order to avoid ambiguity as reviewer suggested and specified that the 2M malic acid solution does not require the addition of NaOH to reach the pH = 1.5 (line 222-223 in the revised manuscript).

→ “The acidities of malic acids (2.0 M) with the various pH values (3.5, 5.5, 7.0, or 8.5) were adjusted using 10 M sodium hydroxide (NaOH). Malic acid with pH 1.5 was made without the addition of NaOH.”

3. Bibliographic citations 4, 6 and 7 should be eliminated because they are superfluous in relation to the article.

: We apologize for our mistake during the revision (Round 1) of references. As the reviewer suggested, we have deleted those references (4, 6, and 7) properly and re-order the references in the revised manuscript.

Reviewer 2 Report

The authors have revised the manuscript according to my comments. I agree to accept it.

Author Response

Journal: Molecules

Manuscript ID: molecules-503337

Title: Structural Characteristics and In Vitro Digestibility of Malic Acid-Treated Corn Starch with Different pH Conditions

Responses to reviewer’s comments (Round 2)

       Reviewer #2

The authors have revised the manuscript according to my comments. I agree to accept it.

: We are very grateful for your decision. Again, we’ll thank you for your review.
